# Automated classification of bacterial cell sub-populations with convolutional neural networks

**Denis Tamiev[1], Paige E. Furman[1], Nigel F. Reuel[2]\***

**1** Department of Biochemistry Biophysics and Molecular Biology, Iowa State University, Ames, Iowa, United States of America, **2** Department of Chemical and Biological Engineering, Iowa State University, Ames, Iowa, United States of America

\* reuel@iastate.edu

## Abstract

Quantification of phenotypic heterogeneity present amongst bacterial cells can be a challenging task. Conventionally, classification and counting of bacteria sub-populations is achieved with manual microscopy, due to the lack of alternative, high-throughput, autonomous approaches. In this work, we apply classification-type convolutional neural networks (cCNN) to classify and enumerate bacterial cell sub-populations (*B. subtilis* clusters). Here, we demonstrate that the accuracy of the cCNN developed in this study can be as high as 86% when trained on a relatively small dataset (81 images). We also developed a new image preprocessing algorithm, specific to fluorescent microscope images, which increases the amount of training data available for the neural network by 72 times. By summing the classified cells together, the algorithm provides a total cell count which is on parity with manual counting, but is 10.2 times more consistent and 3.8 times faster. Finally, this work presents a complete solution framework for those wishing to learn and implement cCNN in their synthetic biology work.

**Data Availability Statement:** Data is available from Figshare (doi: 10.6084/m9.figshare.13100225).

**Funding:** Internal Iowa State University funding.

## Introduction

Control of cell morphology, cycle state, and cell-to-cell interactions are some of the key design goals in synthetic biology to develop 'living materials' [1–4]. As one example, bacteria endospores can be incorporated into materials for revival and functional activation [5]. The phenotypic heterogeneity of cells needs to be controlled for precise functional materials such as sensors or bacterial therapies [6, 7]. Classification of heterogeneity is currently assessed via manual inspection of microscope images, as we did before with *B. subtilis* endospore activated as fluorescent reporters (Fig 1A) [8]. Accurate, higher throughput quantification of cell sub-populations is necessary for improved design of cell-based materials, sensors, and therapies. Such tools would enable better understanding of underlying genetic mechanisms of the cell sub-populations.

Current methods of classification work are insufficient to match the throughput of synthetic design in bacteria. While flow-assisted cell counting (FACS) can address larger,

Competing interests: The authors have declared that no competing interests exist.

**Fig 1. Classifying and counting bacterial sub-populations.** (A) Examples of B. subtilis cells, forespores and spores captured on microscope images, identified manually in prior work. (B) "Raw" or unprocessed fluorescence microscopy image of B. subtilis cells during vegetative growth. (C) Examples of various cell clusters found on fluorescent microscope images (i. Artifact (cell debris, impurity), ii. single cell, iii. four-cell cluster, and iv. ten-cell clusters.). (D) Flow cytometry output fluorescence data for viability assay of B. subtilis cells. (i) Gated region contains signal from live cells. Data presented in log (RFU) scale. (ii) Clusters of data on flow cytometry output from (Di) that likely associate with different cell states (vegetative, spore) but are difficult to determine from the FACS data.

eukaryotic, cells the method cannot be as easily applied to sorting bacterial cells which can be up to 1,000 time smaller than eukaryotes. Moreover, bacteria cells can rapidly transition through development states, such as *B. subtilis* which repeatedly transitions between vegetative cell, forespore, and spore states (Fig 1A) [8]. Moreover, bacteria can form clusters of cells, especially biofilm forming *B. subtilis* (Fig 1B), which further confounds FACS based analysis which can misrepresent a cluster of cells (or cluster of various cell types) as a single cell (Fig 1C) [9]. However, FACS can be used for some bacteria classification where there are clear fluorescent reporters and distinct populations, such as classifying viability of *B. subtilis* (Fig 1Di) [10] using the permeability to green (Syto 9) dye but not red (Propidium Iodide) as an indication of cell viability [11]. However, when a mixture of cells, spores and forespores of *B. subtilis* are subjected to this method more than two clusters of fluorescence signal are present in the flow cytometry data (Fig 1Dii). Correlating these clusters of data to bacterial sub-types can be a challenging task; thus quantification of bacterial cell populations is typically performed with manual microscopy. However, there has been many advances in automated approaches to analysis of microscope images including the use of deep learning.

Deep learning tools such as Convolutional Neural Networks (CNN) are well suited for automating classification and quantification of bacterial phenotypes present in microscope images; they have been applied in other microbiology tasks such as classification of coccoliths formed by various coccolithophores, stalked protozoa identification, and bacterial plankton classification [12–15]. CNN transform an image volume to a linear output volume (holding the class scores) using a stack of interconnected convolutional, pooling, and fully-connected layers. The convolutional and fully-connected layers transform the images using weights and biases that are tuned via gradient descent to match the CNN output (class scores) to the annotated training data set [16]. The architecture of the CNN is based on the number of these layers and how they are stacked (see Supplement 3 in S1 File for summary of CNN framework used in this work and solution steps). A successful demonstration of the capability of modern CNN during the ImageNet Large Scale Visual Recognition Challenge (ILSVRC) in 2012 attracted interest to this field and spiked the development of a variety of more advanced, dense convolutional neural network architectures (DCNN) [17, 18]. The main disadvantage in applying these dense

neural networks is that a large amount of data is required to achieve sufficient performance due to the density of these networks (*e.g.* large number of weights to optimize) [19]. Simpler CNN architectures such as LeNet-5, which has a 7 layer structure, is better suited for simple images and sparse training sets [20, 21]. Fluorescent microscope images are typically single channel (greyscale, unlike 3-channel RGB images) and relative low resolution; also, bacterial cells lack high level features that would necessitate a dense CNN for proper generalization. In addition, simple architectures can require significantly fewer images to achieve proper generalization [22]. One example of a CNN based on LeNet-5, modified to reflect recent progress in CNN design such as implementation of non-saturating non-linearities (ReLU), local response normalization, non-overlapping max-pooling and random dropout layers, was successfully implemented in bacterial colony counting [23–27]. Large colonies of cells present on agar plates have simple circular structure and lack high level features, similar to smaller bacterial cell clusters. Nonetheless, there are unique challenges associated with efficient pre-processing of more limited fluorescent microscope images and adapting CNN to analyze bacterial cell clusters, the scope of this work.

In this work, we demonstrate classification and enumeration of bacterial (*B. subtilis*) sub-populations present in fluorescent microscope images with a classification-type convolutional neural network (cCNN) adapted from the previous, larger colony counting work [23]. To achieve this, we present a more efficient method for microscope image preprocessing and augmentation that improves the training efficiency from a limited set of annotated images. We assess the accuracy and confidence of the algorithm for each sub-population type, and discuss how the quality of the algorithm can be further improved. We benchmark this algorithm against one established tool, ImageJ, for total count of cells. We compare the performance of our cCNN algorithm, in terms of accuracy, speed and repeatability, to manual processing. By archiving all our cCNN code and training data, this work can also serve as a clear example to others wishing to implement cCNN for other classification tasks that are encountered when trying to identify and count the effects of synthetic biology design.

## Results and discussion

### Image preprocessing

While the applications for convolutional neural networks are becoming more widespread and the tools more freely available, these networks are typically designed to work with macroscopic objects with many distinct features (cars, people, animals etc.) [28]. Microorganisms captured on fluorescent microscope images are significantly different in appearance compared to images of common objects (Fig 2A) as they have no real internal features, just the unique size and outline of the cells. For this work we acquired 1,000 fluorescent microscope images (single channel) of *B. subtilis* cells, and evaluated image preprocessing algorithms commonly used with cCNNs. The objective was to identify data preparation methods that resulted in highest performance of common neural networks.

The raw microscope images were segmented using an adaptive binary thresholding algorithm to identify individual clusters of cells that were spatially segregated from each other (Fig 2A, Supplement 2 in S1 File). Some of these clusters were single cells, some were artifacts (dust, debris, etc.) and others were multicellular clusters of cells (nascent biofilm) with a drop off in number as the size of the cluster increased (*i.e.* rarer sub-population, see Supplement 2 in S1 File for population numbers of the segmented data). The varied sizes of the segmented data presented a challenge as the majority of classic neural networks operate based on fixed resolution images. As such, all images must be adjusted to a standard size prior to training the network.

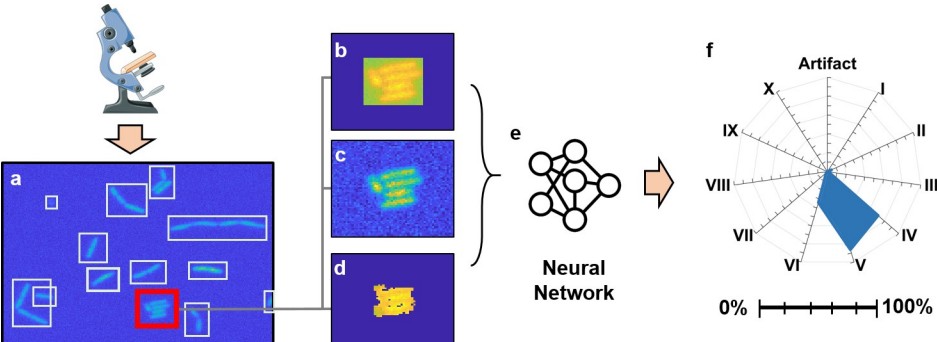

**Fig 2. Image processing workflow.** Raw fluorescent microscope images (a) were processed with a binary segmentation algorithm, and clusters of bacterial cells were manually annotated. All image segments of cell clusters were standardized to the same size with either (b) Null Bumper, (b) Blended or (d) Masked methods. These annotated training images were passed to the cCNN to determine optimal network weights (e). The output of the network (from image depicted in panel c) is a confidence value for each sub-class (A–artifact; I—X–single through ten cell cluster), here presented in a radar chart (F). Major tics– 20% confidence increments, minor– 10%.

Generally, there are three ways to standardize sizing of the image data. Images can be (1) distorted (shrunk or stretched), (2) masked or (3) framed [29]. We chose to standardize cropped images to a 200 by 200-pixel resolution, since most of the cell cluster segments did not exceed these dimensions. Distortion of the cropped images changed the appearance of bacterial cells. Also, application of a mask on cropped images (*i.e.* applying a filter to remove background pixel data) altered the edges of clusters (Fig 2D). Therefore, we settled on framing approaches with two different methods of filling in the frame. In the first framed approach (referred to as "null bumper" or NB), the cropped images were placed in the center of a 200x200 canvas, and the void space was filled with 0 intensity pixels (Fig 2B). In the second, we created a novel "blended bumper" (referred to as BB) where the frames of cropped images were blended with local background pixels, which resulted in a much more consistent appearance of microscope images (Fig 2C, Supplement 2 in S1 File).

The NB and BB image datasets were created by rotating captured images at right angles and in a mirrored dimension, resulting in 8x times as many images. During training of NB and BB networks, we observed that the latter was underfitting (Supplement 5 in S1 File). To reduce the magnitude of underfitting, we trained a third network with images created by rotation at finer angle which generated more training data (referred to as "advanced rotation" or AR). The AR image dataset was created by rotating images by 10 degrees across both dimensions, then blending those images with the same algorithm described for BB, resulting in 72x increase in training images.

As mentioned above, the NB, BB, and AR data were used to train cCNNs (see Supplement 3 in S1 File for code and steps). For this study we selected a network architecture that was demonstrated to work well in a similar classification tasks and used it with all three image datasets [23]. The structure of this cCNN is discussed in Supplement 4 in S1 File. Briefly, the input of the cCNN is an image (2D array of pixel intensities) and the output of the network are confidence values (0%-100%) for each output class (artifact or one, two, three, and so on, through ten cell-cluster). The top confidence was selected to indicate the predicted class (Fig 2F).

To evaluate the quality of the network, annotated images that were not used for training of the network were passed through the network for classification. Accuracy (comparison of prediction to annotated truth value) and confidence (% of how certain the cCNN is of its prediction) were used as metrics to identify best methods of data preparation (NB, BB, *vs*. AR) and overall performance of the network at cell cluster classification tasks.

### Evaluation of networks

Accuracies of CNNs trained on NB, BB, and AR datasets were then evaluated with new images. These results can be tabulated in common confusion matrices (Supplements 7 and 8 in S1 File) that count the precise predictions and number of each off-target predictions. Alternatively, we can visualize this same data using a box and whisker plot (Fig 3), to show the most frequent classifications and the directionality of variance. This plot maps the true (manually annotated count, X- axis) against the algorithm predictions (Y-axis). It is observed that the NB-trained network was much more likely to misclassify A, I and II classes compared to BB- and AR-trained networks. Availability of more training data (AR-trained network), improved the network's performance in classifying III, IV, V-cell clusters.

The average (across all cluster types) accuracies of the blended bumper and null bumper cCNNs were similar, 344% and 40% respectively (Supplement 8 in S1 File). The average accuracy of the AR cCNN was 58%. Upon closer examination, the networks performed best at differentiating between an artifact, single cells and two cell clusters (cells joined together), with accuracies reaching as high as 86%, 73% and 75% for the AR network (Fig 3, Supplement 8 in S1 File).

As another measure of performance, we assess the relative confidences of the NB, BB, and AR trained cCNN on their predictions for each population sub-class (Fig 4). When examining mean confidences for each individual output class, it becomes evident that the NB network classified all cell clusters with lower confidence than BB and AR networks (Fig 4); the average confidences of both BB and AR networks were 12% higher than that of the NB network across all classes (Supplement 9 in S1 File). The most apparent improvement in prediction confidence can be observed for smaller objects (artifacts, single cells and II-cell clusters), and small cell clusters (III-cell cluster). At this small scale, there is a dramatic improvement in network's performance when more data is provided for training using the AR network. We attribute this to the asymmetric features of these smaller clusters that present dramatically new images when rotated. This effect is apparently diminished as the cluster size increases, as the NB and AR

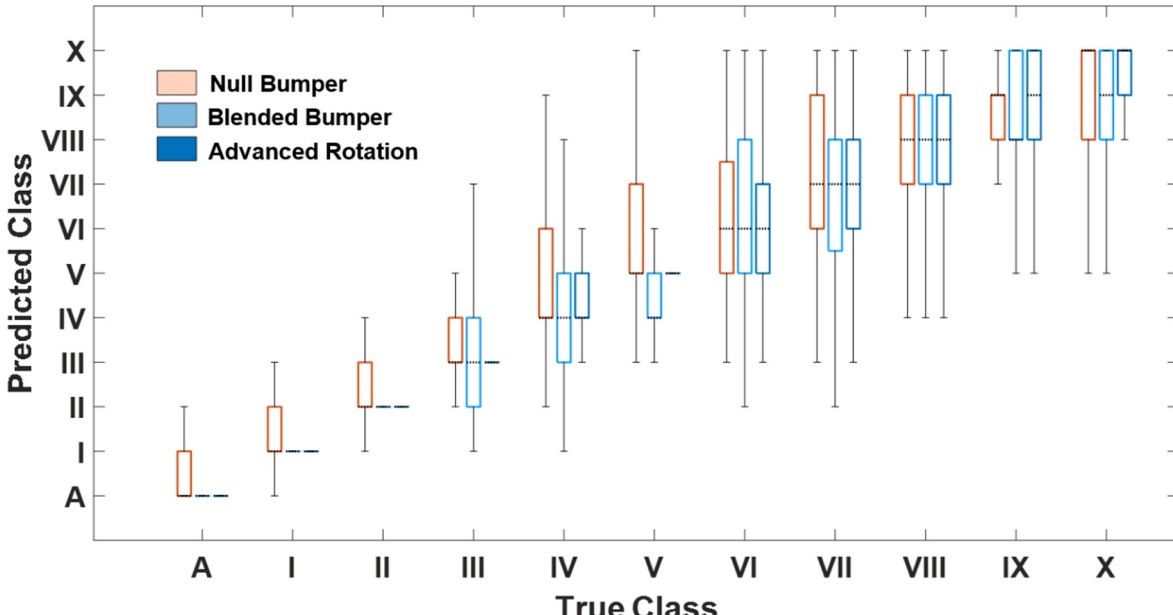

**Fig 3. Accuracy of NB, BB and AR trained cCNNs across all output classes (A, I-X cell clusters).** Standard box and whisker levels are used with the center mark (dotted line) indicating the median, the bottom and top edges of the box indicating the 25th and 75th percentiles, respectively, the whiskers extending to the most extreme data points not considered outliers.

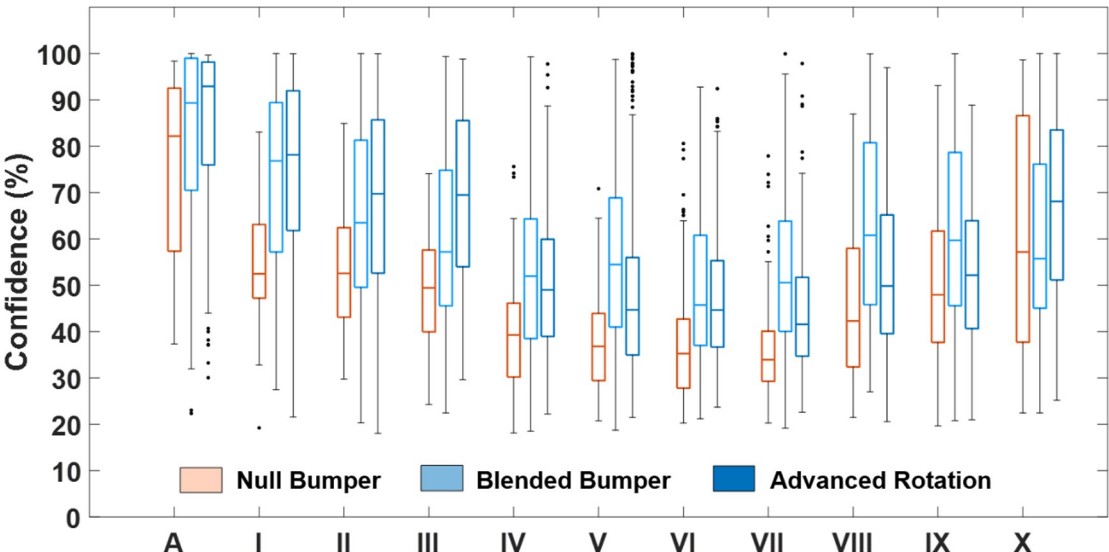

**Fig 4. Confidence of NB, BB and AR trained cCNN across all output classes (A, I-X).** Standard box and whisker levels are used with the center mark indicating the median, the bottom and top edges of the box indicating the 25th and 75th percentiles, respectively, the whiskers extending to the most extreme data points not considered outliers, and the outliers plotted individually.

perform at the same relative level. These larger clusters become more symmetric and thus rotation does not present dramatically new images on which to train.

The apparent increase in performance at the higher cluster count may also be an artifact due to the limitation of classes imposed by the network. There are only 11 outputs available and thus anything larger would get placed in the largest feature class. However, this could also be an effect of more features present on the larger cluster; additional images of large cell clusters would have to be obtained to determine if this is a real effect or artifact.

The variability in accuracy and confidence between different classes can be explained by the fact that the networks were trained on a limited number of training images which did not capture all spatial and rotational orientations of multicellular clusters. These could be further improved using more training data, although at this scale the cCNN approach provides suitable confidence in distinguishing smaller cell clusters and artifacts especially when using the AR training data. Additionally, the variability in sub-population count is less important when each class is summed together to obtain a total count of cells, which is another common application in synthetic biology applications, which we present next.

### Comparing cCNN total cell count to ImageJ

As mentioned in the introduction, total count of bacterial cells, especially those prone to clustering, is difficult to do with FACS and is an important metric in quantifying effects of synthetic design. Here we benchmark our cCNN data (summing counts across classes) against two existing methods: 1) a popular open source tool ImageJ that relies on a watershed algorithm and 2) manual counting [30].

For benchmarking the automated approaches, 24 new microscope images were segmented with a binary thresholding algorithm, and cell clusters were either classified and summed with a cCNN (NB, BB and AR trained networks) or sub-segmented with a watershed algorithm and counted with ImageJ (Supplement 10 in S1 File). For truth data we performed manual counting (one person, averaging three independent counts) and compare the automated findings in a parity chart (Fig 5). In this representation a slope equal to unity would indicate that the software

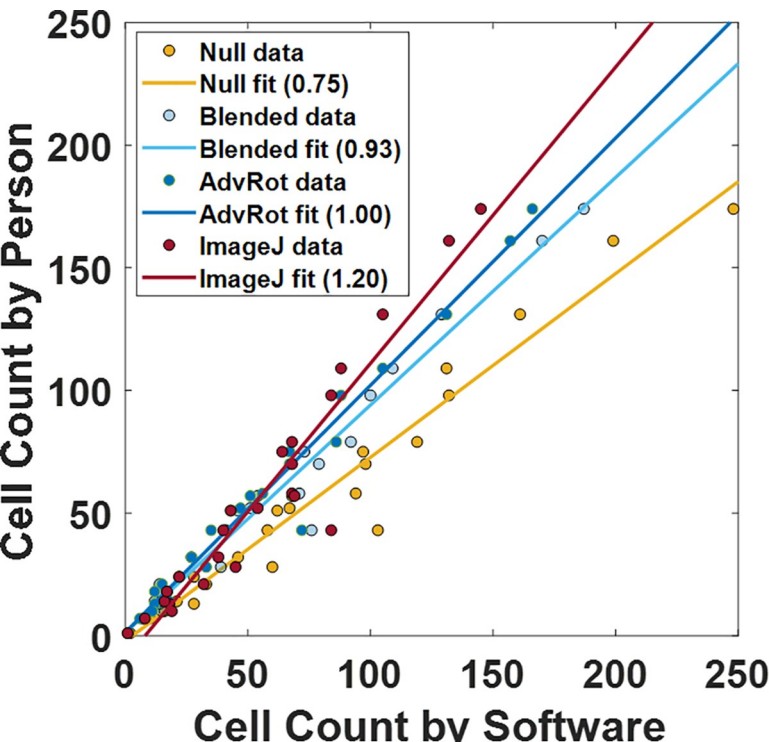

**Fig 5. Comparing accuracy of manual (same person, average of 3 counts) to software (cCNN or ImageJ) counting methods (Supplement 13 in _S1 File_).** Each point represents an evaluated image. The units are the number of cells per image.

approach has the same evaluation as manual counting. NB-trained networks appear to over-count cells on microscope images (slope of 0.75), BB-trained networks perform significantly better (slope of 0.93) but some over counting is still present, and AR-trained networks count equally well as person (slope of 1). ImageJ undercounts images, as evident by the slope of 1.2.

These results clearly demonstrate a significant improvement in average accuracy when comparing NB and BB networks. This improvement is attributed to the quality of the data used to train the network, further supporting that blending the background of images leads to a more accurate network. Some of the neural networks created for this study were overcounting cells on images that contained high cell counts. This can be explained by the fact that high cell count images also contained more multi-cell clusters compared to low cell count images. High cell count images resulted from cell cultures with high cell density, which leads to biofilm formation in *B. subtilis*. As observed previously, the networks developed in this study exhibited lower accuracy at predicting multi-cell clusters, compared to single or two-cell clusters. The undercounting of ImageJ is attributed to its poor capacity to accurately sub-segment images of biofilms with a watershed algorithm (Supplement 11 in S1 File). Surprisingly, while the AR trained network had an average class prediction accuracy of 58%, when these sub-population numbers are summed together the overall count is on par with manual counting.

## Comparing consistency, repeatability, and speed of cCNN total cell count to human performance

While the accuracy of the AR-trained network prepared in this study is similar to that of manual for total cell counting, consistency, repeatability, and speed should vary. To quantify these

metrics and benchmark manual counting against the cCNN approach, five people were recruited to count bacterial cells on printed microscope images that were used for this study. Test subjects were given 25 images, 15 unique and 5 that appeared twice in random order (Supplement 12 in S1 File). The study evaluated two parameters–(1) how the total bacterial cell count on all 25 images compared between five people, and (2) how the cell counts compared when test subjects were presented with duplicate images.

Variability of human test subjects that counted cells on 25 images was, on average, as high as 39.5 cells per image (Fig 6A). As expected, larger variability was observed with images that depicted more cells (Supplement 12 in S1 File). In comparison, variability of the cCNN based algorithm on average did not exceed 3.7 cells per image. Average variability across all images for human test subjects was 10.2 times greater than for cCNN based algorithm (Fig 6A, inset).

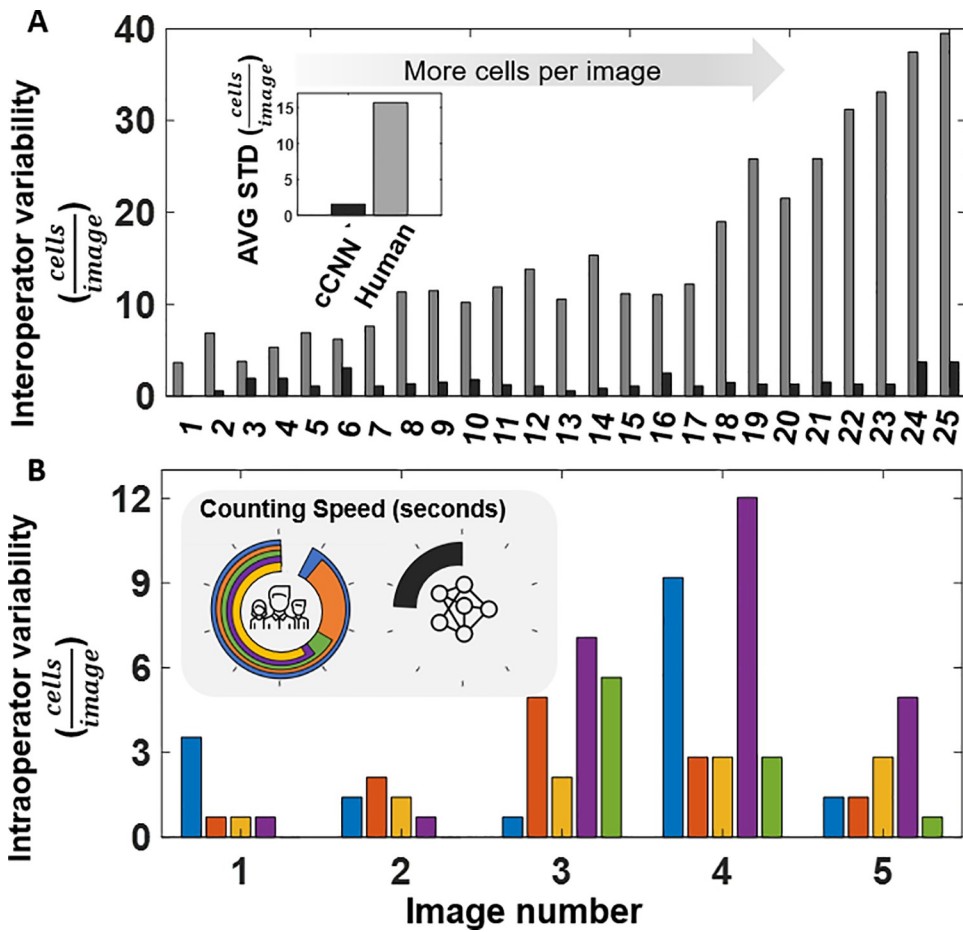

**Fig 6. Comparing consistency and repeatability of manual cell counting.** (A) Standard deviation (STD) as a measure of variance of cell counts across 25 microscope images (ordered in increasing number of cells per image) using manual counting from five separate human counters (interoperator variability) benchmarked against the advanced rotation (AR) trained cCNN algorithm. Average variance of human subjects was 10.2 times larger than that of the cCNN algorithm (insert). (B) Standard deviation of each test subject when counting a duplicate image randomized in their set (intraoperator variability). Images 1 through 5 presented in the order of increasing cell population (Supplement 13 in S1 File). Total time required to count cells on all images for each operator is presented in the pie chart (inset). Full circle is 1000 seconds, tick marks are set to 100 seconds. Each individual is represented with a different color (Blue–Person 1; Orange–Person 2; Yellow–Person 3; Purple–Person 4; Green–Person 5). If the data from a specific person is not present, then their variability was 0 cells/image.

Variability due to human error is not uncommon in manual tasks. In some cases, differences between individuals can be normalized if there are patterns in human behavior. For example, if one person is systematically under or over-counting, a correction factor (bias) can be introduced to correct the results of a cohort. However, it is not always possible to correct results, if the inconsistencies are random. In this study we evaluated inconsistency of specific individuals on duplicate images, and compared it to that of the cCNN based algorithm. Results show that individuals tend to count a different number of cells on the same image that was duplicated (Fig 6B). The difference is reported as standard deviation between the two instances for each individual (Supplement 12 in S1 File). The variability of individuals in some cases was as high as 12.0 cells per image, or 10% (Fig 6B). In contrast, the variability of the algorithm tested on duplicate images was 2.26 cells per image or 2% on average, and it ranged from 0.7 to 4.24, or 1–4% (Supplement 12 in S1 File). It is important to note that the network performs with no variability when it is loaded in the memory of the computer. We traced algorithm variability to the process of loading the network with a computer vision library. It is possible that the global system variables of the device impact how the network is loaded (rounding error).

In addition, we observed that the processing speed of the cCNN based algorithm is considerably faster than that of human test subjects. Specifically, when the algorithm was tested on a computer with an NVIDIA Quadro K620 GPU, the algorithm processed all 25 images up to 3.8 times faster than human test subjects (Supplement 12 in S1 File, Fig 6A inset). Considerably higher processing speeds can be achieved with more computational power.

## Conclusions and summary

Convolutional neural networks have played a significant role in automating many complex image processing tasks in biomedical research such as histology, cellular localization, embryonic cell counting and many others [31–33]. In this paper we explored the prospect of developing a cCNN algorithm that can classify bacteria sub populations (in this case stage of biofilm development) from fluorescent microscope images. This algorithm can then be used to readily quantify the effects of synthetic genetic circuits installed in cell-based materials, therapies, and sensors.

In this paper we tested the classification performance of cCNN trained on three different types of annotated training data sets: null bumper (NB), blended bumper (BB), and advanced rotation (AR). Of these we find that advanced rotation is best suited for improving accuracy and confidence of the CNN for smaller clusters (debris artifacts, single, double, and triple cells) with accuracies of 86%. The larger clusters (4 to 10 cell clusters) would require additional training data to improve the accuracy (now at 50–66%).A significant finding was that although the individual classification accuracies were lower than desired, when the total count of cells is summed, these inconsistencies balance out and near parity is found using our algorithm compared to manual counting when assessed over 24 images. Established algorithms, such as ImageJ, undercounts especially with higher cell count images. We also benchmarked this AR-trained cCNN algorithm against multiple manual counts and found 10.2x reduction in interoperator variability and 3.8X increase in processing speed.

It is important to note that these results were achieved with a modest training data set (annotated segments from 81 images that had representative clusters from 1–10 cells per cluster). Acquiring additional images would be the simplest method to improve the cCNN performance (through additional training) but in this application, this does take significant time with microscopy to search for the rarer, sub-populations (high cell count clusters). Data acquisition of rare events has always been one of the central challenges of the AI industry, but some level of autonomous data acquisition can be achieved with unsupervised learning. Then, this

tool could be used for synthetic biology applications, such as tracking the effect of genetic changes on the timing and heterogeneity of sporulation development stages. We anticipate there will be many other applications for automated classification in this field, and thus have provided all our data and architecture code to serve as another learning example (freely available at http://www.reuelgroup.org/resources.html).

## Methods

### Cell culture prep for microscope imaging

Cells of *B. subtilis* 168 were grown overnight until saturation in LB media using standard methods. The overnight was used as a seed culture to infect sterile LB in a microplate. The growth in the microplate reader was monitored as a measure of absorbance at 600 nm. Cells were grown at 37C, and with reasonable agitation to prevent excess clumping. Cells were harvested at various stages of growth–early growth, log phase, saturation. When necessary, the culture was diluted with fresh LB prior to microscope imaging. More detailed instructions on sample preparation are outlined in the Supplement 1 in S1 File.

### Flow cytometry

Cells were diluted to the appropriate density using LB with 5% sodium azide solution. Then, cell samples were used with the BacLight bacterial viability and Counting Kit (ThermoFisher). All flow cytometry experiments were performed on the factory-direct unmodified BD FACS-Canto flow cytometer (San Jose, CA). We used 488 nm laser for extinction and detected fluorescence with a 525/550 nm as well as 610/620 nm bandpass filters as described previously [8].

### Fluorescent microscopy and image preprocessing

Images of *B. subtilis* cell culture were acquired on the Nikon Eclipse E800 microscope equipped with a FITC filter set. Cells were stained with Ethidium Bromide to promote fluorescence. Raw images (unaltered pixel intensity) were segmented with an adaptive binary thresholding algorithm. Images of individual cells or cell clusters were cropped, saved, and annotated using a custom Matlab GUI (Supplement 13 in S1 File). All cell clusters were examined, and we determined that the image segments did not exceed 200 by 200 pixel resolution. As such, we set the input layer of the neural network to accept images of 200 by 200 pixel resolution (Supplement 3 in S1 File). In the null bumper approach, cropped images of cell clusters were placed in the middle of the 200 by 200 matrix, and the pixels around the cropped image were set to 0 intensity. In the blended bumper approach, the space around the cropped image was filled with pixel intensities that matched those in the local background of the cropped image (Supplement 2 in S1 File).

For this study, we acquired 1,000 fluorescent microscope images, and then manually annotated these (20,855 segments in total) and 10% of the data was used for training the network and the remainder was reserved for evaluation of performance (see Supplement 2 in S1 File for population numbers of the segmented data).

### Convolutional neural network and training

The structure of the convolutional neural network used in this study is described in the Supplement section 3 [23]. Briefly, it is a feed-forward network with 4 convolutional layers. Image data was preprocessed on local computers, while the training procedures were executed with Amazon Web Service resources (Supplement 4 in S1 File).

### Evaluating network

Trained networks were evaluated with reserved data (10% of the original data set). The output of the network from these annotated, evaluation images was scored with a confusion matrix (Supplement 6 in S1 File).

### Counting cells

The networks were also used to perform cell counting. To achieve that, raw fluorescence microscope images were preprocessed with an adaptive binary thresholding algorithm. Cropped images of cells and cell clusters were normalized to the 200 by 200 pixel resolution, and supplied to the classification neural network. The output of the network corresponded to the number of cells in a given cell clusters. The individual outputs were then added to find the total cell count.

For ImageJ adaptive binary thresholding is first used to find clusters and then these cell clusters were further sub-segmented with a watershed algorithm. The image is treated as a topological map, with highest pixel intensities representing the bottom of the "basin". The basins are then filled with pixels until pixels from neighboring basins come in contact. The boundary is drawn in the place of the contact. The total number of sub-segmented cells was then reported as a total cell count.

### Manual cell counting

Fluorescent microscope images that were used to test the efficiency of software-based cell counting were also counted manually. These human cell counting experiments were conducted under the IRB's oversight (IRB ID 19–566) and is described in more detail in the Supplement sections 10 (for Fig 5), and 12 (for Fig 6).

## Supporting information

**S1 File.**
(DOCX)

## Acknowledgments

We would like to thank Shawn Rigby for assistance with acquiring, reviewing and discussing the flow cytometry data.

## Author Contributions

**Conceptualization:** Denis Tamiev, Paige E. Furman, Nigel F. Reuel.

**Data curation:** Denis Tamiev, Paige E. Furman, Nigel F. Reuel.

**Formal analysis:** Denis Tamiev, Paige E. Furman, Nigel F. Reuel.

**Funding acquisition:** Denis Tamiev, Paige E. Furman, Nigel F. Reuel.

**Investigation:** Denis Tamiev, Paige E. Furman, Nigel F. Reuel.

**Methodology:** Denis Tamiev, Paige E. Furman, Nigel F. Reuel.

**Project administration:** Denis Tamiev, Paige E. Furman, Nigel F. Reuel.

**Resources:** Denis Tamiev, Paige E. Furman, Nigel F. Reuel.

**Software:** Denis Tamiev, Paige E. Furman, Nigel F. Reuel.

**Supervision:** Denis Tamiev, Paige E. Furman, Nigel F. Reuel.

**Validation:** Denis Tamiev, Paige E. Furman, Nigel F. Reuel.

**Visualization:** Denis Tamiev, Paige E. Furman, Nigel F. Reuel.

**Writing – original draft:** Denis Tamiev, Paige E. Furman, Nigel F. Reuel.

**Writing – review & editing:** Denis Tamiev, Paige E. Furman, Nigel F. Reuel.

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
