## [Decision Letter · Decision Letter 0]

28 Aug 2020

PONE-D-20-24391

Automated Classification of Bacterial Cell Sub-Populations with Convolutional Neural Networks

PLOS ONE

Dear Dr. Reuel,

Thank you for submitting your manuscript to PLOS ONE. After careful consideration, we feel that it has merit but does not fully meet PLOS ONE’s publication criteria as it currently stands. Therefore, we invite you to submit a revised version of the manuscript that addresses the points raised during the review process.

I agreed with the comments from both reviewers and their decisions that your manuscript requires major revision. Please address their comments by supplementing your current manuscript with additional materials in your revised manuscript in order to be considered for acceptance. 

We look forward to receiving your revised manuscript.

Kind regards,

Yan Chai Hum

Academic Editor

PLOS ONE

Journal Requirements:

Reviewers' comments:

Reviewer's Responses to Questions

**Comments to the Author**

1. Is the manuscript technically sound, and do the data support the conclusions?

Reviewer #1: No

Reviewer #2: Yes

2. Has the statistical analysis been performed appropriately and rigorously? 

Reviewer #1: Yes

Reviewer #2: No

3. Have the authors made all data underlying the findings in their manuscript fully available?

Reviewer #1: No

Reviewer #2: Yes

4. Is the manuscript presented in an intelligible fashion and written in standard English?

Reviewer #1: Yes

Reviewer #2: Yes

5. Review Comments to the Author

Reviewer #1: 1. No details on the computer vision algorithms.

2. Most of the references are outdated.

3. Comparison with previous studies are not available.

4. Summarize your contribution using at least one paragraph to explain the reason why other readers should read your article, what are the significance of this study and what knowledge this article provides?

5. It looks like an application rather than a research to me. Correct me if I am wrong.

Reviewer #2: problem statement and project motivation are clearly defined. However, limited explanation on their proposed technique cCNN, why is this neural network been selected? There are many types of CNN are made available for this kind of similar classification works. No benchmarks literature are available to support their proposed techniques. Presently literature are very limited with only 19 references, many state-of-arts neural networks in recent years are not discussed in the present manuscript.

6. PLOS authors have the option to publish the peer review history of their article (what does this mean?). If published, this will include your full peer review and any attached files.

Reviewer #1: No

Reviewer #2: **Yes: **Khin Wee Lai

---

## [Author Response · Author response to Decision Letter 0]

22 Sep 2020

Thank you for providing us with these details on how to appropriately format the manuscript. We have made the changes to comply with the editing style of PLOS One. Please see the edited manuscript, and let us know if you see anything that requires additional stylistic edits.

2. Is the manuscript technically sound, and do the data support the conclusions?

a. Reviewer #1: No

b. Reviewer #2: Yes

Thank you for providing that feedback. We have strengthened the manuscript through the comments below.

3. Has the statistical analysis been performed appropriately and rigorously? 

a. Reviewer #1: Yes

b. Reviewer #2: No

Thank you for providing that feedback. We have strengthened the manuscript through the comments below.

4. Have the authors made all data underlying the findings in their manuscript fully available?

a. Reviewer #1: No

b. Reviewer #2: Yes

Thank you for providing that feedback. We can see how it can be overlooked that the data (labeled microscope images, and trained neural networks) were made available, as they are located on our group’s website (http://www.reuelgroup.org/resources.html). We reviewed the manuscript, and made sure that the correct links are listed. Additional information, such as Matlab code, is available in the supplement section. If the reviewers feel that additional code or other details need to be included in the main manuscript or the supplement, we will gladly address those requests prior to the publication. 

5. Review Comments to the Author

Reviewer #1: 

1. No details on the computer vision algorithms.

Thank you for outlining this weakness of our paper. We revised the introduction section, and provided additional detail on the computer vision algorithm, the architecture of the neural network, and its origins. If you see any seminal review or publication missing, we would be interested in including.

We described the process of image preprocessing (which is what we defined as computer vision algorithms) in the results section. Lines 103-110 discuss object segmentation. In the following paragraph, lines 111-128, we described data augmentation methods used in this paper. Specifically, we talked about size normalization, and images rotation. The rest of the paragraph was dedicated towards elaborating on the type of the network that was used in this study. Additional information our choice of the neural network was described in the introduction. We decided that the training process information better fits in the supplement section of the paper, and as such the reader was referred to the supplement sections.

With that said, we conceive of the possibility that the reader might have a slightly different definition of what “computer vision algorithms” means, and, perhaps, this is what the reviewer is referring to. If there are additional details you would like us to add to our description of computer vision algorithms used for this paper, please provide specifics and we can make those edits.

2 Most of the references are outdated.

Thank you for expressing your concern about the age of the cited literature. We revisited the reference literature used in our manuscript, and cited recent reviews describing architectures of CNN, and use cases of CNN in microbiology imaging. This was done to also satisfy response to Reviewer #2’s similar request to update the reference literature. Please see below.

3. Comparison with previous studies are not available.

Thank you for pointing this out. We would like to compare our approach to other classification architectures, but we cannot find other studies describing the use of CNN architectures to classify and enumerate cells from fluorescent microscope images. As such, we benchmarked our method with current go-to methods used by biologists such as manual image analysis and feature based recognition, like ImageJ. If there is a specific algorithm or paper to which you want us to benchmark our approach, please point us to the reference. 

4. Summarize your contribution using at least one paragraph to explain the reason why other readers should read your article, what are the significance of this study and what knowledge this article provides?

Our primary target audience consists of cell researchers who routinely use microscopy to classify and enumerate cells present on a slide. Our paper shows how to prepare such images for regression and expand a limited image data set using the rotation algorithm. It then presents a thorough guide of using a simple CNN architecture to determine network weights and classify images. For the cell type presented in this work (B. subtilis) we show superior accuracy and speed than existing methods such as manual counting and feature based recognition (ImageJ). We provide all code, and thus this technique can be readily adopted by others for other cell types, providing more accurate and timely results in their work. Our secondary audience is all those learning to use CNN type architectures in their work. Our paper and accompanying data sets (all labeled images) presents a useful learning module for those wanting to go through the design steps of classification with CNN before applying to their problem.

5. It looks like an application rather than a research to me. Correct me if I am wrong.

This work is a research application. It is true we are not a group that works on new CNN architectures. Instead we are applying an existing architecture to a new problem, one that we (and others to date) have had to rely on manual counting to achieve. In doing so, we had to determine how to expand a limited data set for training and evaluate the performance against existing methods. Many papers that we routinely read are similar in scope (see examples below), using an established technique in a new application area. We posit that this has merit in building the foundation of science, as it expands the utility of a new method/tool. Without such application build out, there never would be dissemination of a new method.1–5 

Examples: 

(1) Derrien, T.; Estellé, J.; Sola, S. M.; Knowles, D. G.; Raineri, E.; Guigó, R.; Ribeca, P. Fast Computation and Applications of Genome Mappability. PLOS ONE 2012, 7 (1), e30377. https://doi.org/10.1371/journal.pone.0030377.

(2) Lu, H.-Y. Application of Optimal Designs to Item Calibration. PLOS ONE 2014, 9 (9), e106747. https://doi.org/10.1371/journal.pone.0106747.

(3) Faria, B.; Abreu, F. V. de. Cellular Frustration Algorithms for Anomaly Detection Applications. PLOS ONE 2019, 14 (7), e0218930. https://doi.org/10.1371/journal.pone.0218930.

(4) Onsorodi, A. H. H.; Korhan, O. Application of a Genetic Algorithm to the Keyboard Layout Problem. PLOS ONE 2020, 15 (1), e0226611. https://doi.org/10.1371/journal.pone.0226611.

(5) Mehta, A. S.; Lau, D. T.-Y.; Wang, M.; Islam, A.; Nasir, B.; Javaid, A.; Poongkunran, M.; Block, T. M. Application of the Doylestown Algorithm for the Early Detection of Hepatocellular Carcinoma. PLOS ONE 2018, 13 (8), e0203149. https://doi.org/10.1371/journal.pone.0203149.

Reviewer #2: 

Problem statement and project motivation are clearly defined. However, limited explanation on their proposed technique cCNN, why is this neural network been selected? There are many types of CNN are made available for this kind of similar classification works. 

Thank you for pointing out this weakness in our manuscript. We agree that a thorough description of the justification for selecting the CNN architecture that we used was lacking. To address, we reworked the introduction section, and included that discussion. 

There are many types of CNN are made available for this kind of similar classification works. No benchmarks literature are available to support their proposed techniques. Presently literature are very limited with only 19 references, many state-of-arts neural networks in recent years are not discussed in the present manuscript.

We greatly appreciate this feedback, and agree that additional references can help the reader gain a better understanding of the background. We revised the introduction to contain a description of the current strategy for selecting networks, as well as a brief history of the field of deep learning. We backed up that information with additional, relevant citations. If we have missed a seminal work, please bring this to our attention, and we will be happy to add it to the list of references.

---

## [Decision Letter · Decision Letter 1]

12 Oct 2020

Automated Classification of Bacterial Cell Sub-Populations with Convolutional Neural Networks

PONE-D-20-24391R1

Dear Dr. Reuel,

We’re pleased to inform you that your manuscript has been judged scientifically suitable for publication and will be formally accepted for publication once it meets all outstanding technical requirements.

Kind regards,

Yan Chai Hum

Academic Editor

PLOS ONE

Additional Editor Comments (optional):

Reviewers' comments:

Reviewer's Responses to Questions

**Comments to the Author**

1. If the authors have adequately addressed your comments raised in a previous round of review and you feel that this manuscript is now acceptable for publication, you may indicate that here to bypass the “Comments to the Author” section, enter your conflict of interest statement in the “Confidential to Editor” section, and submit your "Accept" recommendation.

Reviewer #2: All comments have been addressed

2. Is the manuscript technically sound, and do the data support the conclusions?

Reviewer #2: Yes

3. Has the statistical analysis been performed appropriately and rigorously? 

Reviewer #2: Yes

4. Have the authors made all data underlying the findings in their manuscript fully available?

Reviewer #2: Yes

5. Is the manuscript presented in an intelligible fashion and written in standard English?

Reviewer #2: Yes

6. Review Comments to the Author

Reviewer #2: all comments have been addressed accordingly. I would recommend for acceptance of this article in current form.

7. PLOS authors have the option to publish the peer review history of their article (what does this mean?). If published, this will include your full peer review and any attached files.

Reviewer #2: No

---

## [Editor Report · Acceptance letter]

16 Oct 2020

PONE-D-20-24391R1 

Automated Classification of Bacterial Cell Sub-Populations with Convolutional Neural Networks 

Dear Dr. Reuel:

I'm pleased to inform you that your manuscript has been deemed suitable for publication in PLOS ONE. Congratulations! Your manuscript is now with our production department. 

Kind regards, 

on behalf of

Dr. Yan Chai Hum 

Academic Editor

PLOS ONE